# Robust Hypothesis Testing Using Wasserstein Uncertainty Sets

**Rui Gao**
School of Industrial and Systems Engineering
Georgia Institute of Technology
Atlanta, GA 30332
rgao32@gatech.edu

**Liyan Xie**
School of Industrial and Systems Engineering
Georgia Institute of Technology
Atlanta, GA 30332
lxie49@gatech.edu

**Yao Xie**
School of Industrial and Systems Engineering
Georgia Institute of Technology
Atlanta, GA 30332
yao.xie@isye.gatech.edu

**Huan Xu**
School of Industrial and Systems Engineering
Georgia Institute of Technology
Atlanta, GA 30332
huan.xu@isye.gatech.edu

## Abstract

We develop a novel computationally efficient and general framework for robust hypothesis testing. The new framework features a new way to construct uncertainty sets under the null and the alternative distributions, which are sets centered around the empirical distribution defined via Wasserstein metric, thus our approach is data-driven and free of distributional assumptions. We develop a convex safe approximation of the minimax formulation and show that such approximation renders a nearly-optimal detector among the family of all possible tests. By exploiting the structure of the least favorable distribution, we also develop a tractable reformulation of such approximation, with complexity independent of the dimension of observation space and can be nearly sample-size-independent in general. Real-data example using human activity data demonstrated the excellent performance of the new robust detector.

## 1 Introduction

Hypothesis testing is a fundamental problem in statistics and an essential building block for scientific discovery and many machine learning problems such as anomaly detection. The goal is to develop a decision rule or a detector which can discriminate between two (or multiple) hypotheses based on data and achieve small error probability. For simple hypothesis test, it is well-known from the Neyman-Pearson Lemma that the likelihood ratio between the distributions of the two hypotheses is optimal. However, in practice, when the true distribution deviates from the assumed nominal distribution, the performance of the likelihood ratio detector is no longer optimal and it may perform poorly.

Various robust hypothesis testing frameworks have been developed, to address the issue with distribution misspecification and outliers. The robust detectors are constructed by introducing various uncertainty sets for the distributions under the null and the alternative hypotheses. In non-parametric setting, Huber's original work [13] considers the so-called $\epsilon$-contamination sets, which contain distributions that are close to the nominal distributions in terms of total variation metric. The more recent works [17, 9] consider uncertainty set induced by Kullback-Leibler divergence around a nominal distribution. Based on this, robust detectors usually depend on the so-called least-favorable distributions (LFD). Although there has been much success in theoretical results, *computation* remains a

major challenge in finding robust detectors and finding LFD in general. Existing results are usually only for the one-dimensional setting. In multi-dimensional setting, finding LFD remains an open question in the literature. In parametric setting, [1] provides a computationally efficient and provably near-optimal framework for robust hypothesis testing based on convex optimization.

In this paper, we present a novel computationally efficient framework for developing *data-driven robust minimax detectors* for non-parametric hypothesis testing based on the Wasserstein distance, in which the robust uncertainty set is chosen as all distributions that are close to the empirical distributions in Wasserstein distance. This is very practical since we do not assume any parametric form for the distribution, but rather "let the data speak for itself". Moreover, the Wasserstein distance is a more *flexible* measure of closeness between two distributions. The distance measures used in other non-parametric frameworks [13, 17, 9] are not well-defined for distributions with non-overlapping support, which occurs often in (1) *data-driven* problems, in which we often want to measure the closeness between an empirical distribution and some continuous underlying true distribution, and (2) *high-dimensional* problems, in which we may want to compare two distributions that are of high dimensions but supported on two low-dimensional manifolds with measure-zero intersection.

To solve the minimax robust detector problem, we face at least three difficulties: (i) The hypothesis testing error probability is a nonconvex function of the decision variable; (ii) The optimization over all possible detectors is hard in general since we consider any infinite-dimensional detector with nonlinear dependence on data; (iii) The worst-case distribution over the uncertainty sets is also an infinite dimensional optimization problem in general. To tackle these difficulties, in Section 3, we develop a safe approximation of the minimax formulation by considering a family of tests with a special form that facilitates a convex approximation. We show that such approximation renders a nearly-optimal detector among the family of all possible tests (Theorem 1), and the risk of the optimal detector is closely related to divergence measures (Theorem 2). In Section 4, exploiting the structure of the least favorable distributions yielding from Wasserstein uncertainty sets, we derive a tractable and scalable convex programming reformulation of the safe approximation based on strong duality (Theorem 3). Finally, Section 5 demonstrates the excellent performance of our robust detectors using real-data for human activity detection.

## 2   Problem Set-up and Related Work

Let $\Omega \subset \mathbb{R}^d$ be the observation space where the observed random variable takes its values. Denote by $\mathscr{P}(\Omega)$ be the set of all probability distributions on $\Omega$. Let $\mathcal{P}_1, \mathcal{P}_2 \subset \mathscr{P}(\Omega)$ be our uncertainty sets associated with hypothesis $H_1$ and $H_2$. The uncertainty sets are two families of probability distributions on $\Omega$. We assume that the true probability distribution of the observed random variable belongs to either $\mathcal{P}_1$ or $\mathcal{P}_2$. Given an observation $\omega$ of the random variable, we would like to decide which one of the following hypotheses is true

$$
\begin{aligned}
H_1 : & \quad \omega \sim P_1, \quad P_1 \in \mathcal{P}_1, \\
H_2 : & \quad \omega \sim P_2, \quad P_2 \in \mathcal{P}_2.
\end{aligned}
$$

A *test* for this testing problem is a (Lebesgue) measurable function $T : \Omega \to \{1, 2\}$. Given an observation $\omega \in \Omega$, the test accepts hypotheses $H_{T(\omega)}$ and rejects the other. A test is called *simple*, if $\mathcal{P}_1, \mathcal{P}_2$ are singletons.

The *worst-case risk of a test* is defined as the maximum of the worst-case type-I and type-II errors

$$
\epsilon(T|\mathcal{P}_1, \mathcal{P}_2) := \max \Big( \sup_{P_1 \in \mathcal{P}_1} P_1\{\omega : T(\omega) = 2\}, \ \sup_{P_2 \in \mathcal{P}_2} P_2\{\omega : T(\omega) = 1\} \Big).
$$

Here, without loss of generality, we define the risk to be the maximum of the two types of errors. Our framework can extend directly to the case where the risk is defined as a linear combination of the Type-I and Type-II errors (as usually considered in statistics).

We consider the *minimax robust hypothesis test* formulation, where the goal is to find a test that minimizes the worst-case risk. More specifically, given $\mathcal{P}_1, \mathcal{P}_2$ and $\epsilon > 0$, we would like to find an $\epsilon$-optimal solution of the following problem

$$
\inf_{T:\Omega \to \{1,2\}} \epsilon(T|\mathcal{P}_1, \mathcal{P}_2). \tag{1}
$$

We construct our uncertainty sets $\mathcal{P}_1, \mathcal{P}_2$ to be centered around two empirical distributions and defined using the Wasserstein metric. Given two empirical distributions $Q_k = (1/n_k) \sum_{i=1}^{n_k} \delta_{\widehat{\omega}_i^k}$, which are based on samples drawn from two underlying distributions respectively, where $\delta_\omega$ denotes the Dirac measure on $\omega$. Define the sets using Wasserstein metric (of order 1):

$$\mathcal{P}_k = \{P \in \mathscr{P}(\Omega) : \mathcal{W}(P, Q_k) \leq \theta_k\}, \ k = 1, 2, \tag{2}$$

where $\theta_k > 0$ specifies the radius of the set, and $\mathcal{W}(P, Q)$ denotes the Wasserstein metric of order 1:

$$\mathcal{W}(P, Q) := \min_{\gamma \in \mathscr{P}(\Omega^2)} \left\{ \mathbb{E}_{(\omega, \omega') \sim \gamma} \left[ \|\omega - \omega'\| \right] : \ \gamma \text{ has mariginal distributions } P \text{ and } Q \right\},$$

where $\| \cdot - \cdot \|$ is an arbitrary norm on $\mathbb{R}^n$. We consider Wasserstein metric of order 1 for the ease of exposition. Intuitively, the joint distribution $\gamma$ on the right-hand side of the above equation can be viewed as a transportation plan which transports probability mass from $P$ to $Q$. Thus, the Wasserstein metric between two distributions equals the cheapest cost (measured in some norm $\| \cdot - \cdot \|$) of transporting probability mass from one distribution to the other. In particular, if both $P$ and $Q$ are finite-supported, the above minimization problem reduces to the transportation problem in linear programming. Wasserstein metric has recently become popular in machine learning as a way to measuring the distance between probability distributions, and has been applied to a variety of areas including computer vision [25, 16, 23], generative adversarial networks [2, 10], and distributionally robust optimization [6, 7, 4, 27, 26].

## 2.1 Related Work

We present a brief review on robust hypothesis test and related work. The most commonly seen form of hypothesis test in statistics is simple hypothesis. The so-called simple hypothesis test assuming that the null and the alternative distributions are two singleton sets. Suppose one is interested in discriminating between $H_0 : \theta = \theta_0$ and $H_1 : \theta = \theta_1$, when the data $x$ is assumed to follow a distribution $f_\theta$ with parameter $\theta$. The likelihood ratio test rejects $H_0$ when $f_{\theta_1}(x)/f_{\theta_0}(x)$ exceeds a threshold. The celebrated Neyman-Pearson lemma says that the likelihood ratio is the most powerful test given a significance level. In other words, the likelihood ratio test achieves the minimum Type-II error given any Type-I error. In practice, when the true distributions deviate from the two assumed distributions, especially in the presence of outliers, the likelihood ratio test is no longer optimal. The so-called robust detector aims to extend the simple hypothesis test to composite test, where the null and the alternative hypotheses include a family of distributions. There are two main approaches to the minimax robust hypothesis testing, one dates back to Huber's seminal work [13], and one is attributed to [17]. Huber considers composite hypotheses over the so-called $\epsilon$-contamination sets which are defined as total variation classes of distributions around nominal distribution, while the more recent work [17, 9] considers uncertainty sets defined using the Kullback-Leibler (KL) divergence, and demonstrated various closed-form LFDs for one-dimensional setting. However, in the multi-dimensional setting, there remains the computational challenge to establish robust sequential detection procedures or to find the LFD. Indeed, closed-form LFDs are found only for one-dimensional case (e.g, [12, 18, 9]). Moreover, classic hypothesis test is usually parametric in that the distribution functions under the null and the alternative are assumed to be belong to a family of distributions with certain parameters.

Recent works [8, 14] take a different approach from the classic statistical approach for hypothesis testing. Although "robust hypothesis test" are not mentioned, the formulation therein is essentially minimax robust hypothesis test, when the null and the alternative distributions are parametric with the parameters belong to certain convex sets. They show that when exponential function is used as a convex relaxation, the optimal detector corresponds to the likelihood ratio test between the two LFDs that are solved from a convex programming. Our work is inspired by [8, 14] and extends the state-of-the-art in several ways. First, we consider more general classes of convex relaxations, and show that they can produce a nearly-optimal detector for the original problem and admits an exact tractable reformulation for common convex surrogate loss functions. In contrast, the tractability of the framework in [8] relies heavily on the particular choice of the convex loss, because their parametric framework has stringent convexity requirement in distribution parameters which fails to hold for general convex loss even for Gaussian case, while our non-parametric framework only requires convexity in distribution which holds for general convex surrogates and imposes no conditions on the considered distributions. In addition, certain convex loss functions render a tighter nearly-optimal

detector than the one considered in [8]. Furthermore, the tractability of our framework is due to novel strong duality results Proposition 1 and Theorem 3. They are nontrivial, and to the best of our knowledge, cannot be obtained from extending strong duality results on robust hypothesis testing [8] and distributionally robust optimization (DRO) [4, 6, 7], as will be elaborated later. We finally remark that [24] also considered using Wasserstein metric for hypothesis testing and drew connections between different test statistics. Our focus is different from theirs as we consider Wasserstein metric mainly for the minimax robust formulation.

## 3  Optimal Detector

We consider a family of tests with a special form, which is referred as a *detector*. A detector $\phi : \Omega \to \mathbb{R}$ is a measurable function associated with a test $T_\phi$ which, for a given observation $\omega \in \Omega$, accepts $H_1$ and rejects $H_2$ whenever $\phi(\omega) \geq 0$, otherwise accepts $H_2$ and rejects $H_1$. The restriction of problem (1) on the sets of all detectors is

$$\inf_{\phi:\Omega\to\mathbb{R}} \max \Big( \sup_{P_1\in\mathcal{P}_1} P_1\{\omega : \phi(\omega) < 0\}, \sup_{P_2\in\mathcal{P}_1} P_2\{\omega : \phi(\omega) \geq 0\}\Big). \tag{3}$$

We next develop a safe approximation of problem (3) that provides an upper bound via convex approximations of the indicator function [22]. We introduce a notion called *generating function*.

**Definition 1** (Generating function). *A generating function $\ell : \mathbb{R} \to \mathbb{R}_+ \cup \{\infty\}$ is a nonnegative valued, nondecreasing, convex function satisfying $\ell(0) = 1$ and $\lim_{t\to-\infty} \ell(t) = 0$.*

For any probability distribution $P$, it holds that $P\{\omega : \phi(\omega) < 0\} \leq \mathbb{E}_P[\ell(-\phi(\omega))\}]$. Set

$$\Phi(\phi; P_1, P_2) := \mathbb{E}_{P_1}[\ell \circ (-\phi)(\omega))] + \mathbb{E}_{P_2}[\ell \circ \phi(\omega)].$$

We define the *risk of a detector* for a test $(\mathcal{P}_1, \mathcal{P}_2)$ by

$$\epsilon(\phi|\mathcal{P}_1, \mathcal{P}_2) := \sup_{P_1\in\mathcal{P}_1, P_2\in\mathcal{P}_2} \Phi(\phi; P_1, P_2).$$

It follows that the following problem provides an upper bound of problem (3):

$$\inf_{\phi:\Omega\to\mathbb{R}} \sup_{P_1\in\mathcal{P}_1, P_2\in\mathcal{P}_2} \Phi(\phi; P_1, P_2). \tag{4}$$

We next bound the gap between (4) and (1). To facilitate discussion, we introduce an auxiliary function $\psi$, which is well-defined due to the assumptions on $\ell$:

$$\psi(p) := \min_{t\in\mathbb{R}} [p\ell(t) + (1-p)\ell(-t)], \quad 0 \leq p \leq 1.$$

For various generating functions $\ell$, $\psi$ admits a close-form expression. Table 1 lists some choices of generating function $\ell$ and their corresponding auxiliary function $\psi$. Note that the Hinge loss (last row in the table) leads to the smallest relaxation gap. As we shall see, $\psi$ plays an important role in our analysis, and is closely related to the divergence measure between probability distributions.

Table 1: Generating function (first column) and its corresponding auxiliary function (second column), optimal detector (third column), and detector risk (fourth column).

| $\ell(t)$ | $\psi(p)$ | $\phi^*$ | $1 - 1/2\inf_\phi \Phi(\phi; P_1, P_2)$ |
|---|---|---|---|
| $\exp(t)$ | $2\sqrt{p(1-p)}$ | $\log\sqrt{p_1/p_2}$ | $H^2(P_1, P_2)$ |
| $\log(1 + \exp(t))/\log 2$ | $-(p\log p + (1-p)\log(1-p))/\log 2$ | $\log(p_1/p_2)$ | $JS(P_1, P_2)/\log 2$ |
| $(t+1)_+^2$ | $4p(1-p)$ | $1 - 2\frac{p_1}{p_1+p_2}$ | $\chi^2(P_1, P_2)$ |
| $(t+1)_+$ | $2\min(p, 1-p)$ | $\mathrm{sgn}(p_1 - p_2)$ | $TV(P_1, P_2)$ |

**Theorem 1** (Near-optimality of (4)). *For any distributions $Q_1$ and $Q_2$, and any non-empty uncertainty sets $\mathcal{P}_1$ and $\mathcal{P}_2$, whenever there exists a feasible solution $T$ of problem (1) with objective value less than $\epsilon \in (0, 1/2)$, there exists a feasible solution $\phi$ of problem (4) with objective value less than $\psi(\epsilon)$.*

Theorem 1 guarantees that the approximation (4) of problem (1) is *nearly optimal*, in the sense that whenever the hypotheses $H_1, H_2$ can be decided upon by a test $T$ with risk less than $\epsilon$, there exists a detector $\phi$ with risk less than $\psi(\epsilon)$. It holds regardless the specification of $\mathcal{P}_1$ and $\mathcal{P}_2$. Figure 1 illustrates the value of $\psi(\epsilon)$ when $\epsilon \in (0, 1/2)$.

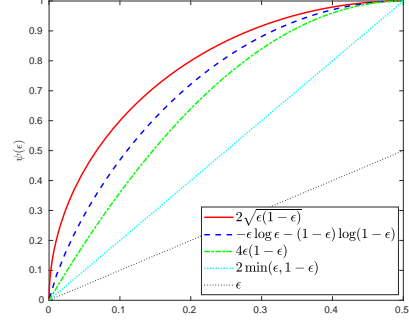

Figure 1: $\psi(\epsilon)$ as a function of $\epsilon$

The next proposition shows that we can interchange the inf and sup operators. Hence, in order to solve (4), we can first solve the problem of finding the best detector for a simple test $(P_1, P_2)$, and then finding the least favorable distribution that maximizes the risk among those best detectors.

**Proposition 1.** *For the Wasserstein uncertainty sets $\mathcal{P}_1, \mathcal{P}_2$ specified in (2), we have*

$$\inf_{\phi:\Omega\to\mathbb{R}} \sup_{P_1\in\mathcal{P}_1, P_2\in\mathcal{P}_2} \Phi(\phi; P_1, P_2) = \sup_{P_1\in\mathcal{P}_1, P_2\in\mathcal{P}_2} \inf_{\phi:\Omega\to\mathbb{R}} \Phi(\phi; P_1, P_2).$$

To establish Proposition 1, observe that the sets under inf and sup are: (i) both infinitely dimensional, (ii) the Wasserstein ball is not compact in the space of probability measures, and (iii) the space of all tests $\phi$ is not endowed with a linear topological structure, so there is no readily applicable tools (such as Sion's minimax theorem used in [8]) to justify the interchange of inf and sup. Our proof strategy is to identify an approximate optimal detector for the sup inf problem on the left side of (5) using Theorem 3 (whose proof does not depend on the result in Proposition 1), and then verify it is also an approximate optimal solution for the original inf sup problem (4). We also note that such issue does not occur in the distributionally robust optimization setting, as their focus is to study only the inner supremum, while the outer infimum in those problems are already finite-dimensional by definition (in fact it corresponds to decision variables in $\mathbb{R}^n$).

The next theorem provides an expression of the optimal detector and its risk.

**Theorem 2** (Optimal detector). *For any distributions $P_1$ and $P_2$, let $\frac{dP_k}{d(P_1+P_2)}$ be the Radon-Nikodym derivative of $P_k$ with respect to $P_1 + P_2$, $k = 1, 2$. Then*

$$\inf_{\phi:\Omega\to\mathbb{R}} \Phi(\phi; P_1, P_2) = \int_\Omega \psi\big(\tfrac{dP_1}{d(P_1+P_2)}\big)d(P_1 + P_2).$$

*Define $\Omega_0(P_1, P_2) := \big\{\omega \in \Omega : 0 < \frac{dP_k}{d(P_1+P_2)}(\omega) < 1, \ k = 1, 2\big\}$. Suppose there exists a well-defined map $t : \Omega \to \mathbb{R}$ such that*

$$t^*(\omega) \in \arg\min_{t\in\mathbb{R}} \Big[ \tfrac{dP_1}{d(P_1+P_2)}(\omega)\ell(-t) + \tfrac{dP_2}{d(P_1+P_2)}(\omega)\ell(t) \Big].$$

*Then $\phi^*(\omega) := -t^*(\omega)$ is an optimal detector for the simple test.*

**Remark 1.** By definition, $\psi(0) = \psi(1) = 0$. Then the infimum depends only on the value of $P_1, P_2$ on $\Omega_0$, the subset of $\Omega$ on which $P_1$ and $P_2$ are absolutely continuous with respect to each other:

$$\inf_{\phi:\Omega\to\mathbb{R}} \Phi(\phi; P_1, P_2) = \int_{\Omega_0} \psi\big(\tfrac{dP_1}{d(P_1+P_2)}\big)d(P_1 + P_2).$$

This is intuitive as we can always find a detector $\phi$ such that its value is arbitrarily close to zero on $\Omega \setminus \Omega_0$. In particular, if $P_1$ and $P_2$ have measure-zero overlap, then $\inf_\phi \Phi(\phi; P_1, P_2)$ equals zero, that is, the optimal test for the simple test $(P_1, P_2)$ has zero risk.

**Optimal detector $\phi^*$.** Set $p_k := dP_k/(d(P_1 + P_2))$ on $\Omega_0$, $k = 1, 2$. For the four choices of $\psi$ listed in Table 1, the optimal detectors $\phi^*$ on $\Omega_0$ are listed in the third column, where sgn denotes the sign function. The first one has been considered in [1].

**Relation between divergence measures and the risk of the optimal detector.** The term $\int_\Omega \psi\big(\frac{dP_1}{d(P_1+P_2)}\big)d(P_1 + P_2)$ can be viewed as a "measure of closeness" between probability distributions. Indeed, in the fourth column of Table 1 we show that the smallest detector risk for a simple test $P_1$ *vs.* $P_2$ equals the negative of some divergence between $P_1$ and $P_2$ up to a constant, where $H$, $JS$, $\Delta$, and $TV$ represent respectively the Hellinger distance [11], Jensen-Shannon divergence [20],

triangle discrimination (symmetric $\chi^2$-divergence) [28], and Total Variation metric [28]. It follows from Theorem 2 that

$$\sup_{P_1 \in \mathcal{P}_1, P_2 \in \mathcal{P}_2} \inf_{\phi: \Omega \to \mathbb{R}} \Phi(\phi; P_1, P_2) = \sup_{P_1 \in \mathcal{P}_1, P_2 \in \mathcal{P}_2} \int_\Omega \psi\big(\tfrac{dP_1}{d(P_1+P_2)}\big) d(P_1 + P_2). \qquad (5)$$

The objective on the right-hand side is concave in $(P_1, P_2)$ since by Theorem 2, it equals to the infimum of linear functions $\Phi(\phi; P_1, P_2)$ of $(P_1, P_2)$. Problem (5) can be interpreted as finding two distributions $P_1^* \in \mathcal{P}_1$ and $P_2^* \in \mathcal{P}_2$ such that the divergence between $P_1^*$ and $P_2^*$ is minimized. This makes sense in that the least favorable distribution $(P_1^*, P_2^*)$ should be as close to each other as possible for the worst-case hypothesis test scenario.

## 4    Tractable Reformulation

In this section, we provide a tractable reformulation of (5) by deriving a novel strong duality result. Recall in our setup, we are given two empirical distributions $Q_k = \frac{1}{n_k} \sum_{i=1}^{n_k} \delta_{\widehat{\omega}_k^i}$, $k = 1, 2$. To unify notation, for $l = 1, \ldots, n_1 + n_2$, we set

$$\omega^l = \begin{cases} \widehat{\omega}_1^l, & 1 \le l \le n_1, \\ \widehat{\omega}_2^{l-n_1}, & n_1 + 1 \le l \le n_1 + n_2, \end{cases}$$

and set $\widehat{\Omega} := \{\omega^l : l = 1, \ldots, n_1 + n_2\}$.

**Theorem 3** (Convex equivalent reformulation). *Problem (5) with $\mathcal{P}_1, \mathcal{P}_2$ specified in (2) can be equivalently reformulated as a finite-dimensional convex program*

$$\max_{\substack{p_1, p_2 \in \mathbb{R}_+^{n_1+n_2} \\ \gamma_1, \gamma_2 \in \mathbb{R}_+^{(n_1+n_2)} \times \mathbb{R}_+^{(n_1+n_2)}}} \sum_{l=1}^{n_1+n_2} (p_1^l + p_2^l) \psi\big(\tfrac{p_1^l}{p_1^l + p_2^l}\big)$$

$$\text{subject to} \quad \sum_{l=1}^{n_1+n_2} \sum_{m=1}^{n_1+n_2} \gamma_k^{lm} \left\| \omega^l - \omega^m \right\| \le \theta_k, \ k = 1, 2,$$

$$\sum_{m=1}^{n_1+n_2} \gamma_1^{lm} = \frac{1}{n_1}, \ 1 \le l \le n_1, \quad \sum_{m=1}^{n_1+n_2} \gamma_1^{lm} = 0, \ n_1 + 1 \le l \le n_1 + n_2,$$

$$\sum_{m=1}^{n_1+n_2} \gamma_2^{lm} = 0, \ 1 \le l \le n_1, \quad \sum_{m=1}^{n_1+n_2} \gamma_2^{lm} = \frac{1}{n_2}, \ n_1 + 1 \le l \le n_1 + n_2,$$

$$\sum_{l=1}^{n_1+n_2} \gamma_k^{lm} = p_k^m, \ 1 \le m \le n_1 + n_2, \ k = 1, 2.$$

$$(6)$$

Theorem 3, combining with Proposition 1, indicates that problem (4) is equivalent to problem (6). We next explain various elements in problem (6).

**Decision variables**. $p_k$ can be identified with a probability distribution on $\widehat{\Omega}$, because $\sum_l p_k^l = \sum_{lm} \gamma_k^{lm} = 1$, and $\gamma_k$ can be viewed as a joint probability distribution on $\widehat{\Omega}^2$ with marginal distributions $Q_k$ and $p_k$. We can eliminate variables $p_1, p_2$ by substituting $p_k$ with $\gamma_k$ using the last constraint, so that $\gamma_1, \gamma_2$ are the only decision variables.

**Objective**. The objective function is identical to the objective function of (5), and thus we are maximizing a concave function of $(p_1, p_2)$. If we substitute $p_k$ with $\gamma_k$, then the objective function is also concave in $(\gamma_1, \gamma_2)$.

**Constraints**. The constraints are all linear. Note that $\omega^l$ are parameters, but not decision variables, thus $\left\| \omega^l - \omega^m \right\|$ can be computed before solving the program. The constraints all together are equivalent to the Wasserstein metric constraints $\mathcal{W}(Q_k, p_k) \le \theta_k$.

**Strong duality**. Problem (6) is a restriction of problem (4) in the sense that they have the same objective but (4) restricts the feasible region to the subset of distributions that are supported on a

subset $\widehat{\Omega}$. Nevertheless, Theorem 3 guarantees that the two problems has the same optimal value, because there exists a least favorable distribution supported on $\widehat{\Omega}$, as explained below.

**Intuition on the reformulation**.

We here provide insights on the structural properties of the least favorable distribution that explain why the reduction in Theorem 3 holds. The complete proof of Theorem 3 can be found in Appendix. Suppose $Q_k = \delta_{\widehat{\omega}_k}$, $k = 1, 2$, $\Omega = \mathbb{R}^d$ and $\psi(p) = 2\sqrt{p(1-p)}$. Note that Wasserstein metric measures the cheapest cost (measured in $\|\cdot - \cdot\|$) of transporting probability mass from one distribution to the other. Thus, based on the discussion in Section 3, the goal of problem (5) is to move (part of) the probability mass on $\hat{\omega}_1$ and $\hat{\omega}_2$ such that the negative divergence between the resulting distributions is maximized. The following three key observations demonstrate *how to move the probability mass in a least favorable way*.

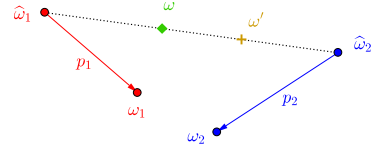

Figure 2: Illustration of the least favorable distribution: it is always better off to move the probability mass from $\widehat{\omega}_1$ and $\widehat{\omega}_2$ to an identical point $\omega$ on the line segment connecting $\widehat{\omega}_1, \widehat{\omega}_2$.

(i) Consider feasible solutions of the form

$$(P_1, P_2) = \big((1-p_1)\delta_{\widehat{\omega}_1} + p_1\delta_{\omega_1},\ (1-p_2)\delta_{\widehat{\omega}_2} + p_2\delta_{\omega_2}\big),\ \ \omega_1, \omega_2 \in \Omega \setminus \{\widehat{\omega}_1, \widehat{\omega}_2\}.$$

Namely, $(P_1, P_2)$ is obtained by moving out probability mass $p_k > 0$ from $\widehat{\omega}_k$ to $\omega_k$, $k = 1, 2$ (see Figure 2). It follows that the objective value

$$\int_\Omega \psi\big(\tfrac{dP_1}{d(P_1+P_2)}\big)d(P_1 + P_2) = \begin{cases} 2\sqrt{p_1 p_2}, & \text{if } \omega_1 = \omega_2, \\ 0, & o.w. \end{cases}$$

This is consistent with Remark 1 in that the objective value vanishes if the supports of $P_1, P_2$ do not overlap. Moreover, when $\omega_1 = \omega_2$, the objective value is independent of their common value $\omega = \omega_1 = \omega_2$. Therefore, we should move probability mass out of resources $\widehat{\omega}_1, \widehat{\omega}_2$ to some common region, which contain points that receive probability mass from both resources.

(ii) Motivated by (i), we consider solutions of the following form

$$(P_1, P_2) = \big((1-p_1)\delta_{\widehat{\omega}_1} + p_1\delta_\omega,\ (1-p_2)\delta_{\widehat{\omega}_2} + p_2\delta_\omega\big),\ \ \omega \in \Omega \setminus \{\widehat{\omega}_1, \widehat{\omega}_2\},$$

which has the same objective value $2\sqrt{p_1 p_2}$. In order to save the budget for the Wasserstein metric constraint, i.e., to minimize the transport distance

$$p_1 \|\omega_1 - \widehat{\omega}_1\| + p_2 \|\omega_2 - \widehat{\omega}_2\|,$$

by triangle inequality we should choose $\omega_1 = \omega_2 = \omega$ to be on the line segment connecting $\widehat{\omega}_1$ and $\widehat{\omega}_2$ (see Figure 2).

(iii) Motivated by (ii), we consider solutions of the following form

$$P_k' = (1 - p_k - p_k')\delta_{\widehat{\omega}_k} + p_1\delta_{\omega_k} + p_1'\delta_{\omega_k'},\ k = 1, 2,$$

where $\omega_k, \omega_k' \notin \Omega \setminus \{\widehat{\omega}_k\}$ are on the line segment connecting $\widehat{\omega}_1$ and $\widehat{\omega}_2$, $k = 1, 2$. Then the objective value is maximized at $\omega_1 = \omega_1' = \widehat{\omega}_2$, $\omega_2 = \omega_2' = \widehat{\omega}_1$, and equals $2\sqrt{(p_1 + p_1')(p_2 + p_2')} + 2\sqrt{(1 - p_1 - p_1')(1 - p_2 - p_2')}$. Hence it is better off to move out probability mass from $\widehat{\omega}_1$ to $\widehat{\omega}_2$ and from $\widehat{\omega}_2$ to $\widehat{\omega}_1$.

Therefore, we conclude that there exist a least favorable distribution supported on $\widehat{\Omega}$. The argument above utilizes Theorem 2, the triangle inequality of a norm and the concavity of the auxiliary function $\psi$. The compete proof can be viewed as a generalization to the infinitesimal setting.

**Complexity**. Problem (6) is a convex program which maximizes a concave function subject to linear constraints. We briefly comment on the complexity of solving (6) in terms of the dimension of the observation space and the sample sizes:

(i) The complexity of (6) is *independent of the dimension* $d$ of $\Omega$, since we only need to compute pairwise distances $\|\omega^l - \omega^m\|$ as an input to the convex program.

(ii) The complexity in terms of the sample sizes $n_1, n_2$ depends on the objective function and can be *nearly sample size-independent* when the objective function is Lipschitz in $\ell_1$ norm (equivalently,

the $\ell_\infty$ norm of the partial derivative is bounded). The reasons are as follows. In this case, after eliminating variables $p_1, p_2$, we end up with a convex program involving only $\gamma_1, \gamma_2$, and the Lipschitz constant of the objective with respect to $\gamma$ is identical to that with respect to $p$. Observe that the feasible region of each $\gamma_k$ is a subset of the $\ell_1$-ball in $\mathbb{R}_+^{(n_1+n_2)}$. Then according to the complexity theory of the first order method for convex optimization [3], when the objective function is Lipschitz in $\ell_1$ norm, the complexity is $O(\ln(n_1) + \ln(n_2))$. Notice that this is true for all except for the first case in Table 1. Hence, this is a quite general.

We finally remark that extending previous strong duality results on DRO [4, 6, 7] from one Wasserstein ball to two Wasserstein balls does not lead to an immediately tractable (convex) reformulation. For one thing, simply applying those previous results on the inner supremum in (4) does not work, because after doing so we are left with the outer infimum that is still intractable. For another thing, applying the previous methodology onto problem (5) does not lead to an tractable reformulation either, mainly because the objective function $\int_\Omega \psi(\frac{dP_1}{d(P_1+P_2)})d(P_1 + P_2)$ is *not separable* in $P_1$ and $P_2$, but depends on the density on the common support of $P_1$ and $P_2$. Thus, as argued in Section 4, in the least-favorable distribution the probability mass of the two distributions cannot be transported arbitrarily, but are *linked* via their common support. In contrast, the problems in DRO [4, 6, 7] have no such linking constraints, which makes it hard to extend the previous methodology. Instead, we prove the strong duality from scratch and provide new insights on the structural properties of the least-favorable distribution that are different in nature from that in DRO settings.

## 5   Numerical Experiments

In this section, we demonstrate the performance of our robust detector using real data for human activity detection. We adopt a dataset released by the Wireless Sensor Data Mining (WISDM) Lab in October 2013. The data in this set were collected with the Actitracker system, which is described in [19, 29, 15]. A large number of users carried an Android-based smartphone while performing various everyday activities. These subjects carried the Android phone in their pocket and were asked to walk, jog, ascend stairs, descend stairs, sit, and stand for specific periods of time.

The data collection was controlled by an application executed on the phone. This application is able to record the user's name, start and stop the data collection, and label the activity being performed. In all cases, the accelerometer data is collected every 50ms, so there are 20 samples per second. There are 2,980,765 recorded time-series in total. The activity recognition task involves mapping time-series accelerometer data to a single physical user activity [29]. Our goal is to detect the change of activity in real-time from sequential observations. Since it is hard to model distributions for various activities, traditional parametric methods do not work well in this case.

For each person, the recorded time-series contains the acceleration of the sensor in three directions. In this setting, every $\omega^l$ is a three-dimensional vector $(a_x^l, a_y^l, a_z^l)$. We set $\theta_1 = \theta_2 = \theta$ as the sample sizes are identical, and $\theta$ is chosen such that the quantity $1 - 1/2 \inf_\phi \Phi(\phi; P_1^*, P_2^*)$ in Table 1, or equivalently, the divergence between $P_1^*$ and $P_2^*$, is close to zero with high probability if $Q_1$ and $Q_2$ are bootstrapped from the data before change, where $P_1^*, P_2^*$ is the LFD yielding from (6). The intuition is that we want the Wasserstein ball to be large enough to avoid false detection while still have separable hypotheses (so the problem is well-defined).

We compare our robust detector, when coupled with CUSUM detector using a scheme similar to [5], with the Hotelling $T^2$ control chart, which is a traditional way to detect the mean and covariance change for the multivariate case. The Hotelling control chart plots the following quantity [21]:

$$T^2 = (x - \mu)'\Sigma^{-1}(x - \mu),$$

where $\mu$ and $\Sigma$ are the sample mean and sample covariance obtained from training data.

As shown in Fig. 3 (a), in many cases, Hotelling $T^2$ fails to detect the change successfully and our method performs pretty well. This is as expected since the change is hard to capture via mean and covariance as Hotelling does.

Moreover, we further test the proposed robust detector, $\phi^* = \frac{1}{2}\ln(p_1^*/p_2^*)$ and $\phi^* = \text{sgn}(p_1^* - p_2^*)$, on 100 sequences of data. Here $p_1^*$ and $p_2^*$ are the LFD computed from the optimization problem (6). For each sequence, we choose the threshold for detection by controlling the type-I error. Then we compare the average detection delay of the robust detector and the Hotelling $T^2$ control chart, as

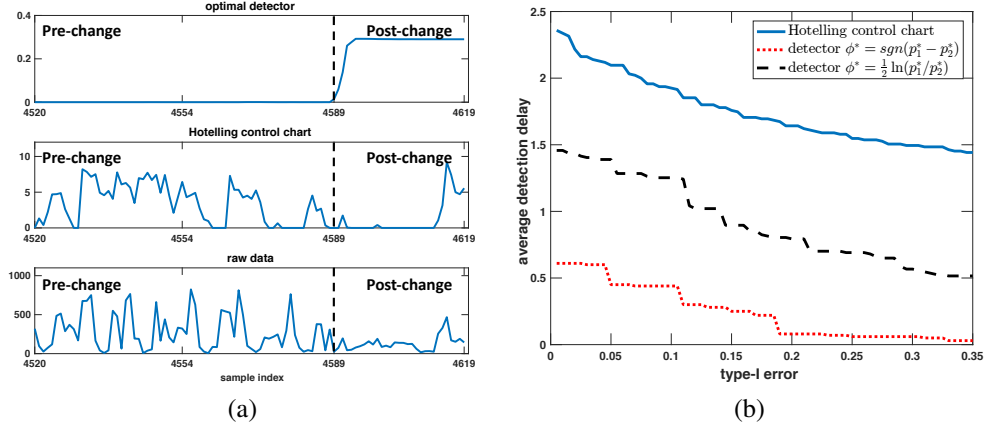

Figure 3: Comparison of the detector $\phi^* = \frac{1}{2}\ln(p_1^*/p_2^*)$ with Hotelling control chart: (a): Upper: the proposed optimal detector; Middle: the Hotelling $\mathrm{T}^2$ control chart; Lower: the raw data, here we plot $(a_x^2 + a_y^2 + a_z^2)^{1/2}$ for simple illustration. The dataset is a portion of full observations from the person indexed by 1679, with the pre-change activity jogging and post-change activity walking. The black dotted line at index 4589 indicates the boundary between the pre-change and post-change regimes. (b): The average detection delay v.s. type-I error. The average is taken over 100 sequences of data.

shown in 3 (b). The robust detector has a clear advantage, and the $\mathrm{sgn}(p_1^* - p_2^*)$ indeed has better performance than $\frac{1}{2}\ln(p_1^*/p_2^*)$, consistent with our theoretical finding.

# 6 Conclusion

In this paper, we propose a data-driven, distribution-free framework for robust hypothesis testing based on Wasserstein metric. We develop a computationally efficient reformulation of the minimax problem which renders a nearly-optimal detector. The framework is readily extended to multiple hypotheses and sequential settings. The approach can also be extended to other settings, such as constraining the Type-I error to be below certain threshold (as the typical statistical test of choosing the size or significance level of the test), or considering minimizing a weighed combination of the Type-I and Type-II errors. In the future, we will study the optimal selection of the size of the uncertainty sets leveraging tools from distributionally robust optimization, and test the performance of our framework on large-scale instances.

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
