[Supplementary Material]

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

# A   Proofs

*Proof of Theorem 1.* Let $T$ be a test with risk $\epsilon(T|\mathscr{P}_1, \mathscr{P}_2) \leq \epsilon$. For $\chi \in \{1, 2\}$, set $\Omega_\chi = \{\omega \in \Omega : T(\omega) = \chi\}$. Then $P_1(\Omega_2) \leq \epsilon$ for any $P_1 \in \mathscr{P}_1$, and $P_2(\Omega_1) \leq \epsilon$ for any $P_2 \in \mathscr{P}_2$. We choose $\phi$ such that

$$
\phi(\omega) = \begin{cases} c_1, & \omega \in \Omega_1, \\ c_2, & \omega \in \Omega_2, \end{cases}
$$

where $c_1 \geq 0 > c_2$. It follows that

$$
\begin{aligned}
& \mathbb{E}_{P_1}[\ell \circ (-\phi)(\omega)] + \mathbb{E}_{P_2}[\ell \circ \phi(\omega)] \\
&= \mathbb{E}_{P_1}\big[\ell \circ (-\phi)(\omega) \cdot \mathbb{1}_{\Omega_1}(\omega)\big] + \mathbb{E}_{P_1}\big[\ell \circ (-\phi)(\omega) \cdot \mathbb{1}_{\Omega_2}(\omega)\big] \\
&\quad + \mathbb{E}_{P_2}\big[\ell \circ \phi(\omega) \cdot \mathbb{1}_{\Omega_1}(\omega)\big] + \mathbb{E}_{P_2}\big[\ell \circ \phi(\omega) \cdot \mathbb{1}_{\Omega_2}(\omega)\big] \\
&= \ell(-c_1) \cdot (1 - P_1(\Omega_2)) + \ell(-c_2) \cdot P_1(\Omega_2) \\
&\quad + \ell(c_1) \cdot P_2(\Omega_1) + \ell(c_2) \cdot (1 - P_2(\Omega_1)) \\
&\leq \ell(-c_1) + \epsilon \cdot (\ell(-c_2) - \ell(-c_1)) + \ell(c_2) + \epsilon(\ell(c_1) - \ell(c_2)) \\
&= \epsilon\ell(c_1) + (1 - \epsilon)\ell(-c_1) + \epsilon\ell(-c_2) + (1 - \epsilon)\ell(c_2).
\end{aligned}
$$

where the inequality follows from $c_2 < c_1$ and the monotonicity of $\ell$. Since $\ell(c_1) > \ell(-c_1)$, $\ell(-c_2) > \ell(c_2)$, and $\epsilon \in (0, \frac{1}{2})$, minimizing over $c_1 \geq 0$ and $c_2 < 0$ yields the result. □

*Proof of Theorem 2.* Note that $P_1, P_2$ are absolutely continuous with respect to $P_1 + P_2$. We have that

$$
\begin{aligned}
& \inf_{\phi:\Omega \to \mathbb{R}} \Phi(\phi; P_1, P_2) \\
&= \inf_{\phi:\Omega \to \mathbb{R}} \int_\Omega \left[\ell(-\phi(\omega))\frac{dP_1}{d(P_1+P_2)}(\omega) + \ell(\phi(\omega))\frac{dP_2}{d(P_1+P_2)}(\omega)\right] d(P_1 + P_2)(\omega)
\end{aligned}
$$

For $\omega$ such that $\frac{dP_1}{d(P_1+P_2)}(\omega) = 1$, by assumptions on the generating function $\ell$, we can define $\phi$ such that $\ell(-\phi(\omega)) \leq \epsilon$ for an arbitrary $\epsilon > 0$. Similarly, for $\omega$ such that $\frac{dP_1}{d(P_1+P_2)}(\omega) = 0$, by assumptions on the generating function $\ell$, we can define $\phi$ such that $\ell(\phi(\omega)) \leq \epsilon$ for an arbitrary $\epsilon > 0$. Hence $\inf_\phi \Phi(\phi; \mu, \nu)$ depends on the integration only on $\Omega_0(P_1, P_2)$. It follows that

$$
\begin{aligned}
& \inf_{\phi:\Omega \to \mathbb{R}} \Phi(\phi; P_1, P_2) \\
&= \inf_{\phi:\Omega \to \mathbb{R}} \int_{\Omega_0} \left[\ell(-\phi(\omega))\frac{dP_1}{d(P_1+P_2)}(\omega) + \ell(\phi(\omega))\frac{dP_2}{d(P_1+P_2)}(\omega)\right] d(P_1 + P_2)(\omega) \\
&= \int_{\Omega_0} \inf_{t \in \mathbb{R}} \left[\ell(-t)\frac{dP_1}{d(P_1+P_2)}(\omega) + \ell(t)\frac{dP_2}{d(P_1+P_2)}(\omega)\right] d(P_1 + P_2)(\omega),
\end{aligned}
$$

where the last inequality follows by interchangeability principle [27]. □

*Proof of Theorem 3.*

*Step 1*. Using Lagrangian and Kantorovich's duality [30], we rewrite the problem as

$$\sup_{\substack{P_1\in\mathcal{P}_1 \\ P_2\in\mathcal{P}_2}} \int_\Omega \psi\big(\tfrac{dP_1}{d(P_1+P_2)}\big)d(P_1+P_2)$$

$$= \sup_{P_1,P_2\in\mathscr{P}(\Omega)} \inf_{\lambda_1,\lambda_2\geq 0} \left\{ \int_\Omega \psi\big(\tfrac{dP_1}{d(P_1+P_2)}\big)d(P_1+P_2) + \lambda_1\theta_1 + \lambda_2\theta_2 \right.$$

$$\left. - \sum_{k=1}^2 \lambda_k \sup_{\substack{u_k\in\mathbb{R}^{n_k} \\ v_k\in L^1(P_k)}} \left\{ \frac{1}{n_k}\sum_{i=1}^{n_k} u_k^i + \int_\Omega v_k dP_k : u_k^i + v_k(\omega) \leq \left\|\omega - \widehat{\omega}_k^i\right\|,\ \forall 1\leq i\leq n_k, \forall\omega\in\Omega \right\} \right\}$$

$$= \sup_{P_1,P_2\in\mathscr{P}(\Omega)} \inf_{\substack{\lambda_1,\lambda_2\geq 0 \\ u_k\in\mathbb{R}^{n_k} \\ v_k\in L^1(P_k)}} \left\{ \int_\Omega \psi\big(\tfrac{dP_1}{d(P_1+P_2)}\big)d(P_1+P_2) + \lambda_1\theta_1 + \lambda_2\theta_2 \right.$$

$$\left. - \sum_{k=1}^2 \lambda_k \left( \frac{1}{n_k}\sum_{i=1}^{n_k} u_k^i + \int_\Omega v_k dP_k \right) : u_k^i + v_k(\omega) \leq \left\|\omega - \widehat{\omega}_k^i\right\|,\ \forall 1\leq i\leq n_k, \forall\omega\in\Omega \right\}.$$

Replacing $\lambda_k u_k^i$ with $u_k^i$ and $\lambda_k v_k$ with $v_k$, the second term and the constraints in the above equation are equivalently written as

$$\sum_{k=1}^2 \left( \frac{1}{n_k}\sum_{i=1}^{n_k} u_k^i + \int_\Omega v_k dP_k \right) : u_k^i + v_k(\omega) \leq \lambda_k \left\|\omega - \widehat{\omega}_k^i\right\|,\ \forall 1\leq i\leq n_k, \forall\omega\in\Omega.$$

Note that such change of variable is valid even when $\lambda_k = 0$. Further note that the objective function is non-increasing in $v_k$, thus we can replace $v_k$ with $\min_{1\leq i\leq n_k}[\lambda_k\left\|\omega - \widehat{\omega}_k^i\right\| - u_k^i]$ without changing the optimal value. Interchanging $\sup$ and $\inf$ yields

$$\sup_{\substack{P_1\in\mathcal{P}_1 \\ P_2\in\mathcal{P}_2}} \int_\Omega \psi\big(\tfrac{dP_1}{d(P_1+P_2)}\big)d(P_1+P_2)$$

$$\leq \inf_{\lambda_1,\lambda_2,u_k,v_k} \left\{ \lambda_1\theta_1 + \lambda_2\theta_2 - \sum_{k=1}^2 \frac{1}{n_k}\sum_{i=1}^{n_k} u_k^i + \right. \tag{7}$$

$$\left. \sup_{P_1,P_2\in\mathscr{P}(\Omega)} \int_\Omega \psi\big(\tfrac{dP_1}{d(P_1+P_2)}\big)d(P_1+P_2) - \int_\Omega \sum_{k=1}^2 v_k dP_k \right\},$$

where the infimum is taken over the set

$$\left\{ \lambda_1,\lambda_2\geq 0, u_k\in\mathbb{R}^{n_k}, v_k(\omega) = \min_{1\leq i\leq n_k}[\lambda_k\left\|\omega - \widehat{\omega}_k^i\right\| - u_k^i],\ \forall\omega\in\Omega,\ k=1,2 \right\}.$$

*Step 2*. We next simplify the inner supremum in (7). We have

$$\sup_{P_1,P_2\in\mathscr{P}(\Omega)} \left\{ \int_\Omega \psi\big(\tfrac{dP_1}{d(P_1+P_2)}\big)d(P_1+P_2) - \sum_{k=1}^2 \int_\Omega \min_{1\leq i\leq n_k}[\lambda_k\left\|\omega - \widehat{\omega}_k^i\right\| - u_k^i]\, dP_k \right\}$$

$$= \max_{\substack{1\leq i_k\leq n_k \\ k=1,2}} \sup_{P_1,P_2} \left\{ \int_\Omega \Big( \psi\big(\tfrac{dP_1}{d(P_1+P_2)}(\omega)\big) - \sum_{k=1}^2 [\lambda_k\left\|\omega - \widehat{\omega}_k^{i_k}\right\| - u_k^{i_k}]\tfrac{dP_k}{d(P_1+P_2)}(\omega) \Big)d(P_1+P_2)(\omega) \right\}$$

For a given solution $(P_1, P_2)$ and for $\omega\in\text{supp}\,(P_1+P_2)$, set

$$T(\omega) = \arg\min_{\omega'\in\Omega} \left\{ \sum_{k=1}^2 [\lambda_k\left\|\omega' - \widehat{\omega}_k^{i_k}\right\| - u_k^{i_k}]\tfrac{dP_k}{d(P_1+P_2)}(\omega) \right\}$$

$$= \begin{cases} \widehat{\omega}_1^{i_1}, & \text{if } \lambda_1\tfrac{dP_1}{d(P_1+P_2)}(\omega) \geq \lambda_2\tfrac{dP_2}{d(P_1+P_2)}(\omega), \\ \widehat{\omega}_2^{i_2}, & o.w. \end{cases}$$

Define another solution $(P_1', P_2')$ such that $P_k'(B) = P_k\{\omega \in \Omega : T(\omega) \in B\}$ for any measurable set $B \subset \Omega$. Then by definition of $T$ we have that

$$\sum_{k=1}^{2} \int_{\Omega} \lambda_k \|\omega_k - \omega\| \, dP_k'(\omega) \leq \sum_{k=1}^{2} \int_{\Omega} \lambda_k \|\omega_k - \omega\| \, dP_k(\omega).$$

In addition, by concavity of $\psi$, for any $p_1, p_1', p_2, p_2' > 0$, it holds that

$$\psi(\tfrac{p_1}{p_1+p_2})(p_1 + p_2) + \psi(\tfrac{p_1'}{p_1'+p_2'})(p_1' + p_2') \leq \psi(\tfrac{p_1+p_1'}{p_1+p_2+p_1'+p_2'})(p_1 + p_2 + p_1' + p_2'),$$

thus it follows that

$$\int_{\Omega} \psi\big(\tfrac{dP_1}{d(P_1+P_2)}\big) d(P_1 + P_2) \leq \int_{\Omega} \psi\big(\tfrac{dP_1'}{d(P_1'+P_2')}\big) d(P_1' + P_2').$$

Hence $(P_1', P_2')$ is a feasible solution that yields an objective value no worse than $(P_1, P_2)$. This suggests that in order to solve the inner supremum of (7), it suffices to only consider $(P_1, P_2)$ such that $\operatorname{supp} P_k \subset \hat{\Omega} := \operatorname{supp} Q_1 \cup \operatorname{supp} Q_2$.

For $l = 1, \ldots, n_1 + n_2$, set $p_l^k = P_k(\omega^l)$, and note that $\gamma_k \in \Gamma(P_k, Q_k)$ can be identified with a non-negative matrix $\gamma_k \in \mathbb{R}_+^{n_1+n_2} \times \mathbb{R}_+^{n_1+n_2}$ with all the column and row sums being 1. Thus, the inner supremum in (7) can now be equivalently written as

$$\sup_{\substack{p_1, p_2 \in \mathbb{R}_+^{n_1+n_2} \\ \sum_l p_k^l = 1}} \left\{ \sum_{1 \leq l \leq n_1+n_2} \psi\big(\tfrac{p_1^l}{p_1^l+p_2^l}\big)(p_1^l + p_2^l) - \sum_{k=1}^{2} \sum_{l=1}^{n_1+n_2} p_k^l \min_{1 \leq i \leq n_k} \cdot [\lambda_k \|\omega^l - \widehat{\omega}_k^i\| - u_k^i] \right\}$$

*Step 3.* It follows from Step 2 that

$$\sup_{\substack{P_1 \in \mathcal{P}_1 \\ P_2 \in \mathcal{P}_2}} \int_{\Omega} \psi\big(\tfrac{dP_1}{d(P_1+P_2)}\big) d(P_1 + P_2)$$

$$\leq \inf_{\lambda_1, \lambda_2 \geq 0} \left\{ \lambda_1 \theta_1 + \lambda_2 \theta_2 + \sup_{\substack{p_1, p_2 \in \mathbb{R}_+^{n_1+n_2} \\ \sum_l p_k^l = 1}} \left\{ \sum_{1 \leq l \leq n_1+n_2} \psi\big(\tfrac{p_1^l}{p_1^l+p_2^l}\big)(p_1^l + p_2^l) \right. \right.$$

$$\left. \left. - \sum_{k=1}^{2} \sum_{l=1}^{n_1+n_2} p_k^l \min_{1 \leq i \leq n_k} \cdot [\lambda_k \|\omega^l - \widehat{\omega}_k^i\| - u_k^i] \right\} \right\}.$$

Applying finite-dimensional convex programming duality on the right-hand side and reverse the procedure in Step 1, we obtain that the right-hand side of (7) is equivalent to (6). Observe that both sides of (7) have the same objective function, but the feasible region of the right-hand side is a subset of that of the left-hand side, and thus the right-hand side should be no greater than the left-hand side, i.e., the above inequality should hold as equality. Thereby we complete the proof.

$\square$

*Proof of Proposition 1.* It suffices to show that

$$\inf_{\phi:\Omega \to \mathbb{R}} \sup_{P_1 \in \mathcal{P}_1, P_2 \in \mathcal{P}_2} \Phi(\phi; P_1, P_2) \leq \sup_{P_1 \in \mathcal{P}_1, P_2 \in \mathcal{P}_2} \int_{\Omega} \psi\big(\tfrac{dP_1}{d(P_1+P_2)}\big) d(P_1 + P_2).$$

Using the strong duality result for distributionally robust optimization with Wasserstein metric [7], $\sup_{P_1 \in \mathcal{P}_1, P_2 \in \mathcal{P}_2} \Phi(\phi; P_1, P_2)$ has an equivalent dual formulation

$$
\sup_{P_1 \in \mathcal{P}_1, P_2 \in \mathcal{P}_2} \Phi(\phi; P_1, P_2)
$$

$$
= \inf_{\substack{\lambda_1, \lambda_2 \geq 0 \\ \pi^1_\# \gamma_k = Q_k}} \left\{ \lambda_1 \theta_1 + \lambda_2 \theta_2 + \int_{\Omega^2} \left[ \ell(-\phi(\omega_1)) - \lambda_1 \|\widehat{\omega}_1 - \omega_1\| \right] \gamma_1(d\widehat{\omega}_1, d\omega_1) \right.
$$

$$
\left. + \int_{\Omega^2} \left[ \ell(\phi(\omega_2)) - \lambda_2 \|\widehat{\omega}_2 - \omega_2\| \right] \gamma_2(d\widehat{\omega}_2, d\omega_2) \right\}
$$

$$
\leq \lambda_1^* \theta_1 + \lambda_2^* \theta_2 + \int_{\Omega^2} \left[ \ell(-\phi(\omega_1)) - \lambda_1^* \|\widehat{\omega}_1 - \omega_1\| \right] \gamma_1^*(d\widehat{\omega}_1, d\omega_1)
$$

$$
+ \int_{\Omega^2} \left[ \ell(\phi(\omega_2)) - \lambda_2^* \|\widehat{\omega}_2 - \omega_2\| \right] \gamma_2^*(d\widehat{\omega}_2, d\omega_2)
$$

$$
= \lambda_1^* \theta_1 + \lambda_2^* \theta_2 + \int_\Omega \left[ \ell(-\phi(\omega)) \frac{d\pi^1_\# \gamma_1^*}{d(\pi^1_\# \gamma_1^* + \pi^1_\# \gamma_2^*)}(\omega) + \ell(\phi(\omega)) \frac{d\pi^1_\# \gamma_2^*}{d(\pi^1_\# \gamma_1^* + \pi^1_\# \gamma_2^*)}(\omega) \right] d(\pi^1_\# \gamma_1^* + \pi^1_\# \gamma_2^*)(\omega)
$$

$$
- \int_{\Omega^2} \lambda_1^* \|\widehat{\omega}_1 - \omega_1\| \gamma_1^*(d\widehat{\omega}_1, d\omega_1) - \int_{\Omega^2} \lambda_2^* \|\widehat{\omega}_2 - \omega_2\| \gamma_2^*(d\widehat{\omega}_2, d\omega_2),
$$

$$(8)$$

where $(\lambda_1^*, \lambda_2^*)$ is dual optimizer of problem (7) associated with the first constraint, $(\gamma_1^*, \gamma_2^*)$ is the optimizer of (7), $\pi^1_\# \gamma_k$ denotes the first marginal distribution of $\gamma_k$, and $\pi^{jk}_\# \gamma$ denotes the marginal distribution of $\gamma$ projected on the $j$-th and the $k$-th coordinates.

On the other hand, by Theorem 3 it holds that

$$
\sup_{P_1 \in \mathcal{P}_1, P_2 \in \mathcal{P}_2} \int_\Omega \psi\left( \frac{dP_1}{d(P_1 + P_2)} \right) d(P_1 + P_2)
$$

$$
= \sup_{P_1 \in \mathcal{P}_1, P_2 \in \mathcal{P}_2} \int_\Omega \inf_{t \in \mathbb{R}} \left[ \ell(-t) \frac{dP_1}{d(P_1 + P_2)}(\omega) + \ell(t) \frac{dP_2}{d(P_1 + P_2)}(\omega) \right] d(P_1 + P_2)(\omega)
$$

$$
= \lambda_1^* \theta_1 + \lambda_2^* \theta_2 + \int_\Omega \inf_{t \in \mathbb{R}} \left[ \ell(-t) \frac{d\pi^1_\# \gamma_1^*}{d(\pi^1_\# \gamma_1^* + \pi^1_\# \gamma_2^*)}(\omega) + \ell(t) \frac{d\pi^1_\# \gamma_2^*}{d(\pi^1_\# \gamma_1^* + \pi^1_\# \gamma_2^*)}(\omega) \right] d(\pi^1_\# \gamma_1^* + \pi^1_\# \gamma_2^*)(\omega)
$$

$$
- \int_{\Omega^2} \lambda_1^* \|\widehat{\omega}_1 - \omega_1\| \gamma_1^*(d\widehat{\omega}_1, d\omega_1) - \int_{\Omega^2} \lambda_2^* \|\widehat{\omega}_2 - \omega_2\| \gamma_2^*(d\widehat{\omega}_2, d\omega_2).
$$

Comparing with (8), we have that for any $\epsilon > 0$, there exists a detector $\phi_\epsilon$, such that

$$
\sup_{P_1 \in \mathcal{P}_1, P_2 \in \mathcal{P}_2} \Phi(\phi_\epsilon; P_1, P_2) \leq \sup_{P_1 \in \mathcal{P}_1, P_2 \in \mathcal{P}_2} \int_\Omega \psi\left( \frac{dP_1}{d(P_1 + P_2)} \right) d(P_1 + P_2) + \epsilon.
$$

Let $\epsilon \to 0$ yields the result.

$$\square$$