[Reviews · NeurIPS 2018]

Reviewer 1



The rebuttal addressed my technical concerns, and also I seemed to have misjudged the size of the contributions at first. My score has been updated. This paper studies the two-sample non-parametric hypothesis testing problem. Given two collections of probability distribution, the paper studies approximating the best detector against the worst distributions from both collections. A standard surrogate loss approximation is used to upper bound the worst case risk (the maximum of the type I and type II errors) with a convex surrogate function, which is known to yield a good solution. Some parts of section 3 are novel, but I'm not familiar enough with the literature to pinpoint exactly what. The convex surrogate approximation is a standard relaxation in learning theory. I wish the authors be more clear in this manner. The main novel contribution seems to be in the setting where the two collections of distributions are wasserstein balls centered around empirical distributions. It that case, Theorem 3 derives a convex relaxation to solving the optimal detector. The authors then provide experimental justification for their method, showing improved performance on real data from previous algorithms. The writing was generally good, though I found the high level ideas hard to follow. Please be more clear about which results apply in which setting. Also, the authors really need to justify why considering wasserstein balls around empirical distributions is a good idea. In fact, I have a few technical concerns: 1. What if the two empirical distributions from P1 and P2 have little overlap? It seems that the hypothesis testing problem becomes trivial 2. Is prop 1 only for the Wasserstein-ball sets? The proof certainly needs this assumption but it's not in the proposition statement. 3. Why in the objective for problem (6) concave? What assumptions on \phi are needed? 4. Please justify the min-max swap in 374 To summarize, given my technical concerns and the fact that the main contribution seems to be deriving a convex relaxation, I'm a bit hesitant to recommend this paper.

Reviewer 2



This paper considers the problem of "minimax robust hypothesis testing" where we want to decide between two composite hypotheses, each of which is a set of distributions. We want a test with low error (maximum of type I and type II) under the worst-case choice of a distribution from each hypothesis set. (The choice of distribution is unknown to the test.) This paper specifically considers the case where we are given two empirical distributions Q_1, Q_2 (on R^n), and we define P_1, P_2 to be balls in the Wasserstein metric centered at Q_1, Q_2. The goal is then to construct a test to decide between the composite hypotheses P_1 and P_2 (given a single fresh sample). The authors use a convex relaxation of the problem in order to give an efficiently-computable test and to prove that it is close to optimal among all possible tests. They also include experiments showing that their method works well on real data. This paper improves upon prior work in a number of ways. Some prior work has considered variants of this problem, but does not give an efficient way to compute the robust detector (except in one dimension). Some further prior work (by which the current work is inspired) does give an efficient detector, but only for the case where the distributions are parametric with parameters belonging to certain convex sets. The current work, on the other hand, is non-parametric and data-driven, imposing no conditions on the distributions whatsoever. I think this is a strong paper. It is well-written and contains a substantial and novel result, making fundamental progress on the question of robust hypothesis testing. EDIT: I have read the other reviews and the author feedback. My opinion of the paper remains the same -- I vote to accept it.

Reviewer 3



This manuscript studied the problem of robust hypothesis testing, i.e., the minimax hypothesis testing between two hypothesis classes which are assumed to be Wasserstein-1 balls around the respective empirical distributions. To tackle this problem, the authors first applied the idea of using convex optimization to solve hypothesis testing (i.e., replace the 0-1 loss by a convex surrogate loss), and then write the dual form of the resulting convex optimization problem to arrive at a finite-dimensional convex optimization problem, which becomes tractable. In other words, the authors provided an efficient and approximate approach to solve the robust hypothesis testing problem. Conceptually speaking, the ideas used in both steps are not new: 1. Replace 0-1 loss by convex surrogate loss for hypothesis testing: this idea has been appeared before in GJN'15 ([8] cited in the manuscript). Although only the exponential function was considered at that time, the extension to general convex functions is quite straightforward. 2. Use strong duality to reduce the infinite-dimensional Wasserstein ball into a finite-dimensional convex program: Jose Blanchet had a series of papers on distributionally robust optimization which covers exactly the same idea and lots of related tools. Although the fact that both ideas are not new may be the main weakness of the current manuscript, in my opinion, it is still really nice in the sense that this manuscript combined both ideas to solve the robust hypothesis testing problem, which is still novel to the best of my knowledge. This combination leads directly to an efficient way to solve the robust hypothesis testing problem, which has considerable applications in practice such as change point detection. This approach also sheds light on solving general minimax hypothesis testing problems efficiently (where the Wasserstein balls can be changed). Edit: I appreciate the authors' feedback and would like to vote for accepting it. I also increased my score from 6 to 7.